# Advanced Convolutional Neural Network-Based Hybrid Acoustic Models for Low-Resource Speech Recognition

**Tessfu Geteye Fantaye** [1] ⓘ**, Junqing Yu** [1,2,*] **and Tulu Tilahun Hailu** [1] ⓘ

1    School of Computer Science & Technology, Huazhong University of Science & Technology,
     Wuhan 430074, China; tessfug@hust.edu.cn (T.G.F.); tutilacs@hust.edu.cn (T.T.H.)
2    Center of Network & Computation, Huazhong University of Science & Technology, Wuhan 430074, China
*    Correspondence: yjqing@mail.hust.edu.cn

**Abstract:** Deep neural networks (DNNs) have shown a great achievement in acoustic modeling for speech recognition task. Of these networks, convolutional neural network (CNN) is an effective network for representing the local properties of the speech formants. However, CNN is not suitable for modeling the long-term context dependencies between speech signal frames. Recently, the recurrent neural networks (RNNs) have shown great abilities for modeling long-term context dependencies. However, the performance of RNNs is not good for low-resource speech recognition tasks, and is even worse than the conventional feed-forward neural networks. Moreover, these networks often overfit severely on the training corpus in the low-resource speech recognition tasks. This paper presents the results of our contributions to combine CNN and conventional RNN with gate, highway, and residual networks to reduce the above problems. The optimal neural network structures and training strategies for the proposed neural network models are explored. Experiments were conducted on the Amharic and Chaha datasets, as well as on the limited language packages (10-h) of the benchmark datasets released under the Intelligence Advanced Research Projects Activity (IARPA) Babel Program. The proposed neural network models achieve 0.1–42.79% relative performance improvements over their corresponding feed-forward DNN, CNN, bidirectional RNN (BRNN), or bidirectional gated recurrent unit (BGRU) baselines across six language collections. These approaches are promising candidates for developing better performance acoustic models for low-resource speech recognition tasks.

**Keywords:** speech recognition; low-resource languages; acoustic models; neural network models

## 1. Introduction

Neural network-based deep learning techniques are the state-of-the-art acoustic modeling approaches by replacing the conventional Gaussian mixture model (GMM) technique since 2011. Then, various researchers have applied those approaches in either hybrid or end-to-end acoustic modeling for developing speech recognition systems. In hybrid acoustic modeling, the acoustic model is developed based on frame-wise cross-entropy or discriminative classification loose functions, which require pre-generated and aligned frame labels using the conventional GMM-hidden Markov model (GMM-HMM) paradigm. In end-to-end acoustic modeling, the acoustic model is built based on the Connectionist temporal classification (CTC) or encoder–decoder objective function, which is used to infer speech-label alignments automatically.

In hybrid acoustic modeling, various deep neural network models, such as feed-forward networks (fully connected deep neural network (DNN) [1–8] and convolutional neural network (CNN) [9–16]) and

deep recurrent neural networks (conventional recurrent neural network (RNN) [17], long short-term memory (LSTM) [18–22], and gated recurrent unit (GRU) [23–26]) are examined for both low- and high-resource-languages speech recognition tasks. These neural network models have strong and weak sides. Of these models, DNN has good discriminative power for classifying the features into the target classes. However, it has the following limitations: (1) It does not have structures that explicitly model the prior knowledge of the speech signals, such as the local properties within the speech frames and long-term dependencies among speech frames. (2) It often severely overfits on the training datasets in low-resource speech recognition, and this results in poor performance of the model on the testing datasets. (3) It does not consider the speech signal frequency variations, which occur due to the speaking system, speaking rate, and speaker emotion. To overcome these limitations, various model regularization techniques (e.g., pretraining algorithms; dropout; activation functions that produce constant gradients including ReLU, maxout, and pnorm; layer and batch normalizations; and L-regularizers, early stopping, and early realignment) are used to make the DNN effective for low-resource speech recognition. Moreover, the speech signal variations in the DNN model can be handle using different speaker adaptive techniques, such as ivector and feature-space maximum likelihood linear regression (fMLLR). Even if the above model size regularization methods reduce the shortcomings of the DNN to some extent, the total number of parameters of the DNN is large, which makes it difficult to train an optimal model for low-resource speech recognition systems.

The CNN is an alternative feed-forward neural network model and is an effective network for representing the local properties within the speech signals spectrum, known as formants [12]. It also considers the frequency variation within the speech signals. Consequently, several researchers have used CNN model for both high-resource [9,15] and low-resource [10–12,14,16] languages speech recognition tasks, and outperformed the DNN models. However, CNN captures only the contexts of the higher layer units, while the information cannot modulate the units in the lower layers. Thus, it is not capable of modeling the long-term context dependencies among speech frames.

The limitations of DNN and CNN models can be overcome using the recurrent neural network models, which have the ability to model long-term dependencies between speech features through unfolding for long time steps. However, the conventional RNN model suffers from vanishing and exploding gradients in the stochastic gradient decent (SGD) training process [19,20,22]. These limitations of RNN model can be reduced using advanced recurrent neural network models (LSTM [12,19,22] and GRU [24,25]). This is because these models are better in modeling the long-term dependencies for speech sequences. However, they suffer from overfitting problem in speech recognition of low-resource languages. Very recently, the gating concept [27,28], highway connection [29–31], and residual connection [32–34] have been shown to be effective for overcoming the gradients vanishing and exploding problems and model overfitting, and for modeling the long-term dependencies among the speech signals for both feed-forward and recurrent neural network models.

Moreover, several advanced neural network acoustic models have been developed through the combination of various feed-forward and recurrent neural network models for acquiring the best quality of the individual models using four different model combination paradigms. First, train the individual neural network acoustic models independently, and then combine the outputs such as a lattice-based system combination approach [12,20]. Second, train the individual neural network acoustic models separately, and then combine these using the combiner layer [35]. Third, train diverse neural network acoustic models by stacking together within a single neural network acoustic model [36–38]. Fourth, train various neural network acoustic models within a single model by integrating them [28,39,40]. The first three model combination paradigms have increased the total number of model parameters, while the fourth can reduce the total model size and save computation resources, which is effective for low-resource speech recognition systems.

The existing neural network acoustic models, including feed-forward and recurrent models, are largely better for speech recognition of high-resource languages than the low-resource languages. Hence, the development of new optimal neural network acoustic models for low-resource-languages

speech recognition systems is an open research area. In this paper, by overcoming the limitations of modeling long-term feature dependencies of the DNN and CNN models, as well as overfitting and gradient vanishing challenges of RNN models, we propose several new optimal advanced convolutional neural network acoustic models by integrating CNN, conventional RNN, and various network connections such as gate, recurrent, highway, and residual via the fourth model combination paradigm for low-resource-language speech recognition systems.

The main contributions of this paper are two folds:

- We propose the gated convolutional neural network (GCNN), recurrent convolutional neural network (RCNN), gated recurrent convolutional neural network (GRCNN), highway recurrent convolutional neural network (HRCNN), residual recurrent convolutional neural network (Res-RCNN), and residual recurrent gated convolutional neural network (Res-RGCNN) models for low-resource-languages speech recognition. These models have not been explored for speech recognition tasks before as far as we know.
- Empirical results using Amharic and Chaha languages and the limited language packages (10-h) of the four Babel languages (Cebuano, Kazakh, Telugu, and Tok-Pisin) speech recognition tasks demonstrate that our models are able to achieve 0.1–42.79% relative performance improvements over the baseline neural network models, namely, DNN, CNN, BRNN, and BGRU models.

The remainder of the paper is organized as follows. Section 2 briefly reviews the state-of-the-art acoustic models of low-resource speech recognition systems. Section 3 describes the proposed neural network models. We report our experimental results in detail in Section 4, including experimental setup, hyperparameters evaluation, and comparisons between selected datasets. Section 5 presents the discussion of various acoustic models. Finally, conclusions and future works are outlined in Section 6.

## 2. State-of-the-Art Acoustic Models

Deep learning methods are the state-of-the-art acoustic modeling approaches in speech recognition of both high-resource and low-resource languages. These approaches include different feed-forward, recurrent, and data-sharing neural network models. Several studies have been conducted on using these methods to develop speech recognition systems for various low-resource languages, as shown in Table 1. The feed-forward acoustic method includes DNN, CNN, and TDNN models. These models are applied for developing speech recognition systems for several low-resource languages. The DNN model is an old feed-forward network that has been used by several researchers in low-resource speech recognitions. For example, Fantaye et al. [7], Sercu et al. [34], Sriranjani et al. [8], and Chan W. and Lane I. [16] used DNN model for Amharic, Cebuano, Hindi, and Bengali languages, respectively. Their findings confirm that DNN has better performance than the conventional GMM models. A CNN model is another alternative feed-forward method that can handle the frequency shifts that are common in speech signals, improve the model robustness, and reduce the model overfitting challenge for speech recognition of low-resource languages. As a result, several studies [12–14,16,34,41] have used CNN in a hybrid CNN-HMM system for low-resource languages. For instance, Sainath and Parada [41] and Huang et al. [13] explored CNN-HMM models for low footprint keyword spotting and speech recognition systems. Cai et al. [14] and Cai and Liu [12] investigated maxout CNN-HMM models for Babel (Bengali, Cantonese, Pashto, Turkish, Tagalog, Vietnamese, and Tamil) languages. The authors compared their results with DNN model and confirmed that CNN is outperformed the DNN models in low-resource speech recognition systems. TDNN is also a feed-forward method which is better than the DNN and CNN models for modeling the long-term contexts of input speech features. It has been used by several researchers for speech recognition of low-resource and very low-resource languages. For instance, Fantaye et al. [42,43] developed rounded phone acoustic modeling units-based and syllable units-based TDNN models for Chaha language. Kang et al. [27] built TDNN models for Swahili and Tamil languages and acquired an absolute word error rate improvement of 0.5 and 0.7

over the DNN models. The results indicate that TDNN is better in terms of recognition performance and convergence speed than DNN and CNN.

Recurrent neural networks are cyclic networks with self-connections from the previous time steps used as inputs to the current time steps. These networks are better to model long-term dependencies among frames of input features. Various studies [12,20,27,44–46] have used the different recurrent neural networks, including conventional RNN, LSTM, and GRU in either hybrid or end-to-end ways for speech recognition of low-resource languages. For example, Cai and Liu [12] investigated the recurrent maxout neural network model for hybrid speech recognition of Babel option period one languages. Kang et al. [20] used the LSTM model to build hybrid speech recognition systems for Pashto, Vietnamese, Swahili, Kazakh, and Tamil languages and achieved an absolute performance improvement of 0.7–5.3 over the DNN models. The authors also explored various new advanced recurrent neural networks, such as local window-BLSTM, local window residual BLSTM, local window BGRU, and local window residual BGRU models, and obtained an absolute performance improvement of 1.3–8.8 over the DNN and LSTM models. Rosenberg et al. [45] and Daneshvar and Veisi [46] also explored GRU encoder–decoder with attention and LSTM-Connectionist Temporal Classification (CTC) end-to-end networks for Babel option period three languages, as well as Persian language speech recognition systems. Their findings show that the recurrent neural network models give better performance than the corresponding feed-forward models. However, these models are easily overfit in speech recognition of low-resource languages.

**Table 1.** Summary of review of the state-of-the-art acoustic models for some low-resource languages.

| Remark | Language | Training Dataset Size (Hours) | Studies | Word Error Rate (WER) (%) | | | | | | |
|---|---|---|---|---|---|---|---|---|---|---|
| | | | | DNN | CNN/Very Deep CNN | TDNN | RNN | LSTM | GRU | End-to-End |
| Ethiopic languages | Chaha | 8.01 | [42] | | | 23.50 | | | | |
| | | | [43] | | | 28.05 | | | | |
| | | | Own | 23.70 | 22.75 | | | 25.82 | 22.36 | |
| | Amharic | 26 | [7] | 11.28 | | | | | | |
| | | | Own | 11.35 | 10.30 | | | 12.76 | 9.89 | |
| European Union language | Lithuanian | 52 | [44] | | | | | | | 1.303 |
| Babel option period three languages | Cebuano | 10.37 | [34] | 76.3 | 74.2/70.3 | | | | | |
| | | | Own | 68.33 | 67.23 | | 72.63 | | 66.41 | |
| | Kazakh | 39.6 | [20] | 54.1 | | | | 52.9 | | |
| | | 9.92 | [34] | 77.3 | 75.2/71.1 | | | | | |
| | | | [47] | 76.2 | | | | | | |
| | | | Own | 70.75 | 70.08 | | 74.85 | | 69.38 | |
| | Telugu | 10.21 | [34] | 87.00 | 85.4/82.5 | | | | | |
| | | | Own | 86.20 | 86.37 | | 89.7 | | 84.98 | |
| | Tok-Pisin | 9.83 | [34] | 62.6 | 59.4/54.2 | | | | | |
| | | | [48] | 52.7 | | | | | | |
| | | | Own | 50.14 | 49.35 | | 54.71 | | 48.22 | |
| | Swahili | 44 | [20] | 46.2 | | | | 42.5 | | |
| | | | [27] | 42.9 | | 42.4 | | 42.1 | | |
| Indian Languages | Tamil | 22 | [8] | 20.01 | | | | | | |
| | Hindi | 22 | [8] | 3.91 | | | | | | |
| Babel option period one languages | Bengali | 10 | [16] | 70.8 | 69.2 | | | | | |
| | Cantonese | 140.7 | [12] | 44.8 | 46.1 | | | 40.7 | | |
| | Pashto | 77.3 | | 51.2 | 52.0 | | | 50.5 | | |
| | | | [20] | 51.2 | | | | 50.5 | | |
| | Turkish | 76.3 | | 47.6 | 47.6 | | | 47.4 | | |
| | Tagalog | 83.7 | [12] | 49.8 | 49.6 | | | 47.9 | | |
| | Vietnamese | 87.1 | | 53.1 | 51.8 | | | 47.8 | | |
| | | | [20] | 53.1 | | | | 47.8 | | |
| | Tamil | 62.3 | [12] | 66.7 | 66.0 | | | 65.0 | | |
| | | | [20] | 66.7 | | | | 65.0 | | |
| Iranian language | Persian | 15 | [46] | | | | | 21.86 | | 17.55 |

Data-sharing networks are either feed-forward networks or recurrent neural networks that are used for transferring the training corpora from high-resource languages to train speech recognition systems for low-resource languages. These methods include multitask learning, multilingual, and weight-transfer approaches. These approaches are the most effective methods for developing

low-resource speech recognition systems. However, our paper scope is far from these methods; hence, we do not review the studies conducted on these models for several low-resource languages.

## 3. Proposed Neural Network Approaches

In this section, we describe the proposed neural network approaches for speech recognition of low-resource languages.

### 3.1. Gated Convolutional Neural Network Model

The CNN model is suitable for modeling the local features by handling the signal variations, but it is unable to model the long-term feature dependencies. This limitation is overcome by using a gate operation, which is widely used in RNNs such as LSTM and GRU. It is helpful to monitor the flow of information between the hidden layers [38] and to model the long-term temporal dependencies of the speech signals [27]. Hence, we combine the gate operation with the CNN layers using an element-wise multiplication operation to develop the GCNN model (Figure 1). The main component of our GCNN model is the gate convolutional layer (GCL), which is defined as:

$$y(x)_k = (y_{k-1} * N + d) \odot \sigma(y_{k-1} * M + e) \tag{1}$$

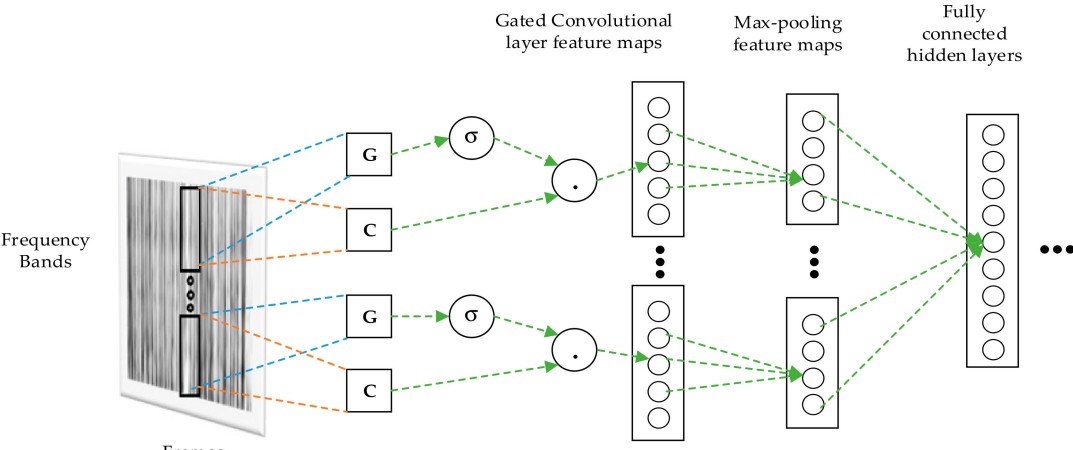

**Figure 1.** Components of the gated convolution neural network model.

The first part of Equation (1) is the standard CNN layer, and the second part is the gate operation. $y_k$ is the output features at $k$, where $k$ = 1, 2, 3, . . . , K, and $N \in R^{a \times b \times c}$ and $M \in R^{a \times b \times c}$ are the convolutional weight matrices. $a$ is the input dimension, $b$ is the size of the convolution filter, and $c$ is the number of filters. The values $d \in R$ and $e \in R$ are the bias of filters. The operators $*$ and $\odot$ are convolutional and element-wise multiplication operations, respectively. The symbol $\sigma$ is a sigmoid nonlinearity. In Figure 1, G and C are the Gate and Convolution, respectively.

Our proposed GCNN model consists of several stacked GCLs interleaved with fully connected DNN or BGRU layers, and then a softmax layer. For every GCL, the max pooling, layer normalization, ReLU nonlinearity, and dropout regularization techniques, successively, are applied for mapping the input feature maps to their corresponding output feature maps. The model configuration of GCNN is described in Table 2. In previous studies, authors have examined this model for different tasks that have static and dynamic datasets, such as handwritten recognition [49], text classification [50], language modeling [51], and speech recognition [27,38] tasks. Wang et al. [38] and Kang et al. [27] investigated the GCNN model for speech command recognition and small-scale speech recognition tasks, respectively. However, our proposed GCNN model is different from these two studies in four ways. First, we use the convolution along the frequency domain using one-dimensional CNN. Second, we stack several fully connected DNN and BGRU layers for better classification. Third, we use the

model regularization methods (layer normalization and dropout) for GCLs to monitor the model overfitting challenge. Fourth, we apply batch normalization to regularize the fully connected DNN and BGRU layers of the GCNN model.

**Table 2.** The configuration of the proposed GCNN, RCNN, and Res-RCNN acoustic models.

| Model | GCNN | RCNN | Res-RCNN |
|---|---|---|---|
| Input map size | $11 \times 40$ Filterbank | $11 \times 40$ Filterbank | $11 \times 40$ Filterbank |
| # layers | 3 GCLs | 3 RCLs | 3 Res-RCLs |
|  | GRCL (3, 60) | RCL (3, 60) | Res-RCL (3, 60) |
| 60 feature maps | Max pooling (3) | Max pooling (3) | Max pooling (3) |
|  | Layer-Normalization ReLU | Layer-Normalization ReLU | Layer-Normalization ReLU |
|  | Dropout (0.15) | Dropout (0.15) | Dropout (0.15) |
|  | GRCL (3, 60) | RCL (3, 60) | Res-RCL (3, 60) |
|  | Max pooling (2) | Max pooling (2) | Max pooling (2) |
| 60 feature maps | Layer-normalization | Layer-normalization | Layer-normalization |
|  | ReLU | ReLU | ReLU |
|  | Dropout (0.15) | Dropout (0.15) | Dropout (0.15) |
|  | GRCL (3, 60) | RCL (3, 60) | Res-RCL (3, 60) |
| 60 feature maps | Layer-Normalization | Layer-Normalization | Layer-Normalization |
|  | ReLU | ReLU | ReLU |
|  | Dropout (0.15) | Dropout (0.15) | Dropout (0.15) |
| 4 Fully connected layers (1024 nodes, batch-normalization, dropout (0.15)) or 3 BGRU layers (550 nodes, batch-normalization, dropout (0.2)) Softmax output layer | | | |

## 3.2. Recurrent Convolutional Neural Network Model

The RCNN model was proposed through the integration of recurrent connection to the CNN model. This model overcomes the limitations of long-term temporal dependencies modeling among feature maps of CNN model, and overfitting and gradient vanishing and exploding problems of RNN models in low-resource speech recognition systems. The recurrent connection is helpful in capturing a wide range of contextual information without depending on the context window. Because of this, RCNN can model the long-term temporal dependencies of the speech frames using the recurrent connection, and the short-term feature dependencies using CNN. The main module of our RCNN model is a recurrent convolutional layer (RCL), whose state evolves over discrete time steps. The RCL is defined as:

$$h_t = f(w * x_t + s * h_{t-1} + b) \tag{2}$$

where $x_t$ and $h_t$ are input feature maps and a hidden state of the RCNN at time step t, respectively. $w$ and $s$ are forward and recurrent convolutional weights, respectively. In addition, $b$ and $f$ represent bias and ReLU nonlinearity.

Several RCLs are stacked together and interleaved with other layers, such as pooling layers, fully connected DNN or BGRU layers, and softmax layer, thereby generating the deep RCNN model. Figure 2 presents the general architecture of the proposed RCL and RCNN model. Table 2 also presents the detailed configuration of RCNN model. Conversely, some studies have investigated this model for computer vision tasks, including image classification [52], text classification [53], object recognition and detection [54], and scene labeling [55] tasks. Inspired by these tasks, Zhao et al. [39] and Tran et al. [40] studied the RCNN for phone recognition and low-resource speech recognition systems, respectively. However, our proposed RCNN model is different from those studies in two ways. First, our RCNN model uses one-dimensional input feature maps, while those studies use two-dimensional input features. Second, the RCLs of our model contains layer normalization and dropout regularization techniques for monitoring the overfitting problems.

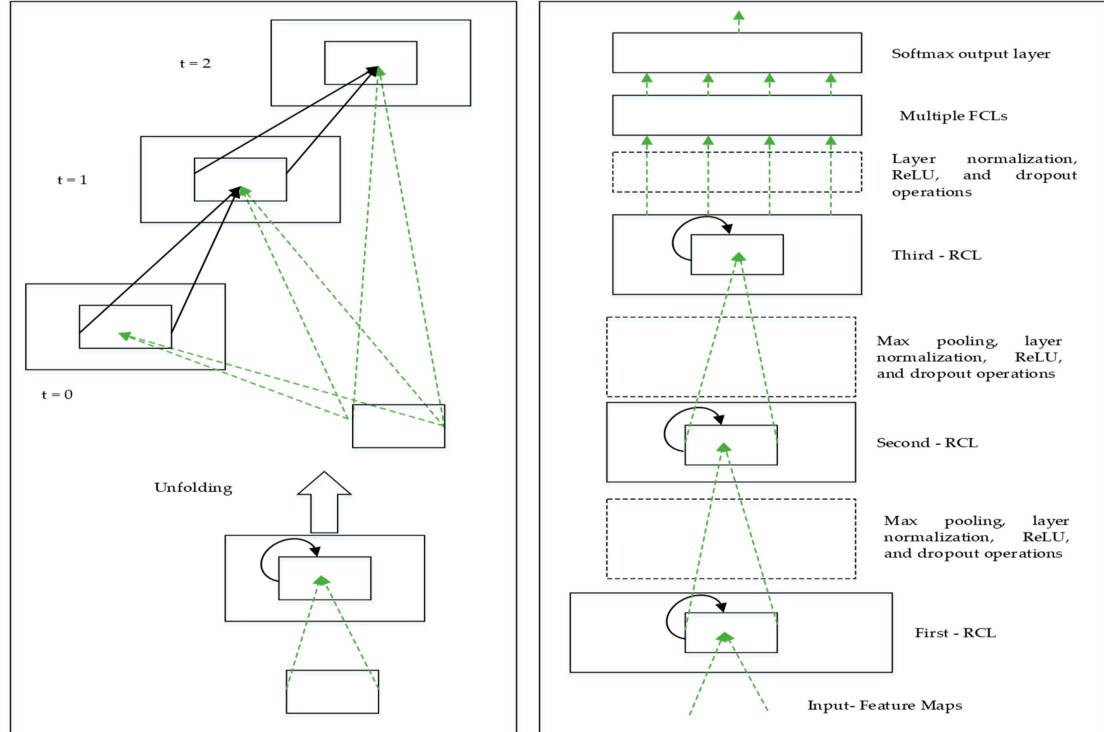

**Figure 2.** (left) RCL with three time-steps; and (right) the architecture of our RCNN model.

### 3.3. Gated Recurrent Convolutional Neural Network Model

Even if the RCNN model represents the long-term temporal dependencies of speech frames better than the baseline CNN model, it might be exposed to vanishing or exploding gradient problems in the SGD training process, similar to the conventional RNN model. Besides, in the recurrent convolutional computation, unrelated context information comes from the recurrent connection, which influences the performance of the RCNN model. To mitigate the above shortcomings, we propose a GRCNN model by integrating a gate operation with the RCNN model. The gate operation in GRCNN helps to control the flow of information in the forward and recurrent connections for modeling the long-term temporal dependencies by minimizing the vanishing and exploding gradients challenges, and it also has the capability to select the most relevant feature frames and inhibits the others. The basic module of our GRCNN model is the gated recurrent convolutional layer (GRCL). Each GRCL contains gate operation, which is defined in Equation (3) as:

$$g(t) = \begin{cases} 0 & \text{if } t = 0 \\ \sigma\left(\left(w_g^f * u(t)\right) + \left(w_g^r * x(t-1)\right)\right) & \text{if } t > 0 \end{cases} \tag{3}$$

where $w_g^f$ represents the feed-forward convolution weights of gate operation, $w_g^r$ denotes the recurrent convolutional weights of the gate operation. The recurrent convolutional weights are shared over all time steps. The symbol $\sigma$ represents a sigmoid nonlinearity. Each GRCL is defined in Equation (4) as:

$$x(t) = \begin{cases} h\left[w^f * u(t)\right] & \text{if } t = 0 \\ h\left[\left(w^f * u(t)\right) + \left(\left(w^r * x(t-1)\right) \odot G(t)\right)\right] & \text{if } t > 0 \end{cases} \tag{4}$$

where $w^f$ and $w^r$ stand for feed-forward and recurrent convolution weights, respectively. Note that h denotes a nonlinear function, ReLU, and $\odot$ is an element-wise multiplication operation. It is assumed that the input to GRCL is the same over time t, which is denoted by $u(0)$. This assumption means that the feed-forward part contributes equally at each time step. It is important to clarify that the time step

in GRCL is not identical to the time associated with the sequential speech feature frames. The time steps denote the iterations in processing the input features. The configuration of our proposed GRCL is presented in Figure 3.

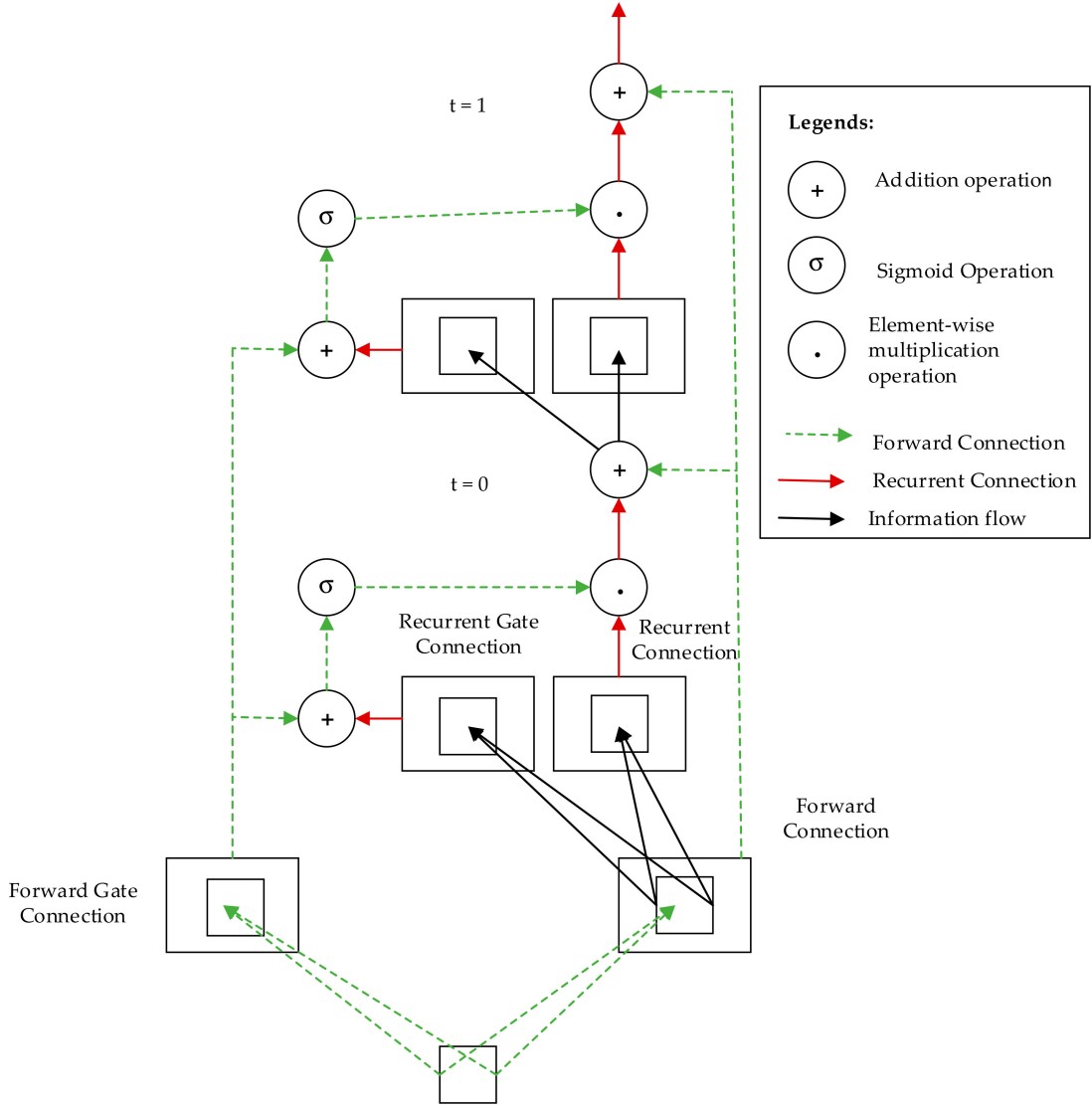

**Figure 3.** Gated recurrent convolutional network (GRCL).

Our GRCNN model consists of several GRCLs, fully connected DNN or BGRU layers, and a softmax output layer. For each of the GRCLs, max pooling, layer normalization, ReLU nonlinearity, and dropout operations are applied, successively, while we use batch normalization for fully connected DNN and BGRU layers, as shown in Table 3.

**Table 3.** The configuration of the proposed baseline CNN and proposed GRCNN, HRCNN, and Res-RGCNN acoustic models.

| Model | CNN | GRCNN | HRCNN | Res-RGCNN |
|---|---|---|---|---|
| Input map size | 11 × 40 Filterbank | 11 × 40 Filterbank | 11 × 40 Filterbank | 11 × 40 Filterbank |
| # layers | 2 conv. layers | 3 GRCLs | 3 HRCLs | 3 Res-RGCLs |
| | conv. (3, 80) | GRCL (3, 80) | HRCL (3, 80) | Res-RGCL (3, 80) |
| | Max pooling (3) | Max pooling (3) | Max pooling (3) | Max pooling (3) |
| 80 feature maps | Layer-normalization | Layer-normalization | Layer-normalization | Layer-normalization |
| | ReLU | ReLU | ReLU | ReLU |
| | Dropout(0.15) | Dropout (0.15) | Dropout (0.15) | Dropout (0.15) |
| | Conv. (3, 80) | GRCL (3, 80) | HRCL (3, 80) | Res-RGCL (3, 80) |
| | Layer-normalization | Max pooling (2) | Max pooling (2) | Max pooling (2) |
| 80 feature maps | ReLU | Layer-normalization | Layer-normalization | Layer-normalization |
| | Dropout(0.15) | ReLU | ReLU | ReLU |
| | | Dropout (0.15) | Dropout (0.15) | Dropout (0.15) |
| | | GRCL (3, 80) | HRCL (3, 80) | Res-RGCL (3, 80) |
| 80 feature maps | | Layer-normalization | Layer-normalization | Layer-normalization |
| | | ReLU | ReLU | ReLU |
| | | Dropout (0.15) | Dropout (0.15) | Dropout (0.15) |
| | 4 Fully connected layers (1024 nodes, batch-normalization, dropout (0.15)) or 3 BGRU layers (550 nodes, batch-normalization, dropout (0.2)) Softmax output layer | | | |

### 3.4. Highway Recurrent Convolutional Neural Network Model

The highway network connection is helpful to introduce phases along which information can flow across several layers without speech signal attenuation [29], to balance the information flow in the forward and recurrent connections in RNN models [30], to increase the depth of the network [29,30,37], and to reduce overfitting and gradient vanishing and exploding effects in the SGD training process of the models [37] when integrating with feed-forward and RNN models. The highway network connection contains two gate operations. A carry gate, C(x), controls the amount of information flow from a lower layer to a higher layer and a transform gate, T(x), controls the amount of information read from the current layer. The highway connection [56] is defined in Equation (5):

$$y = h(x, w_h) \odot T(x, w_t) + x \odot C(x, w_c) \tag{5}$$

where $h(x, w_h)$ represents the neural networks such as feed-forward, RNNs, and other networks. $T(x, w_t)$, $C(x, w_c)$, and $\odot$, respectively, represent transform gate, carry gate, and element-wise multiplication operation.

Conversely, our proposed RCNN model, which is presented in Section 3.2, has its own shortcomings. First, the RCNN model is exposed to the vanishing or exploding gradients problem in the SGD training process. Second, the recurrent convolutional computation of RCNN produces unrelated context information, which comes from the recurrent connection that influences the performance of the RCNN model.

We propose the HRCNN model through the integration of the highway connection with the RCNN model for overcoming the shortcomings of the RCNN model. The basic component of our HRCNN model is the highway recurrent convolutional layer (HRCL), of which the generic architecture is presented in Figure 4, and it is defined in Equations (6)–(8):

$$x(t) = h\left[\left(\left(w^f * u(t)\right) + \left(w^r * x(t-1)\right)\right) \odot T(t) + \left((u(t) + x(t-1)) \odot C(t)\right)\right] \tag{6}$$

$$T(t) = \sigma\left[\left(w_t^f * u(t)\right) + \left(w_t^r * x(t-1)\right)\right] \tag{7}$$

$$C(t) = \sigma\left[\left(w_c^f * u(t)\right) + \left(w_c^r * x(t-1)\right)\right] \tag{8}$$

where $w^f$ and $w^r$ represent the forward and recurrent convolutional weights. $*$ and $\odot$ represent convolution and element-wise multiplication operations, respectively. $h$ denotes the nonlinear function, ReLU, and $x$ is an input feature frame maps. $T(t)$ is the transformation gate operation and $C(t)$ is the carry gate operation. A carry gate is defined as $C(t) = 1 - T(t)$. Hence, Equations (6) and (8) are simplified in Equations (9) and (11), respectively.

$$x(t) = h\Big[\Big(\big(w^f * u(t)\big) + \big(w^r * x(t-1)\big)\Big) \odot T(t) + \big((u(t) + x(t-1)) \odot (1 - T(t))\big)\Big] \tag{9}$$

$$T(t) = \sigma\Big[\Big(w_t^f * u(t)\Big) + \Big(w_t^r * x(t-1)\Big)\Big] \tag{10}$$

$$C(t) = 1 - \sigma\Big[\Big(w_t^f * u(t)\Big) + \Big(w_t^r * x(t-1)\Big)\Big] \tag{11}$$

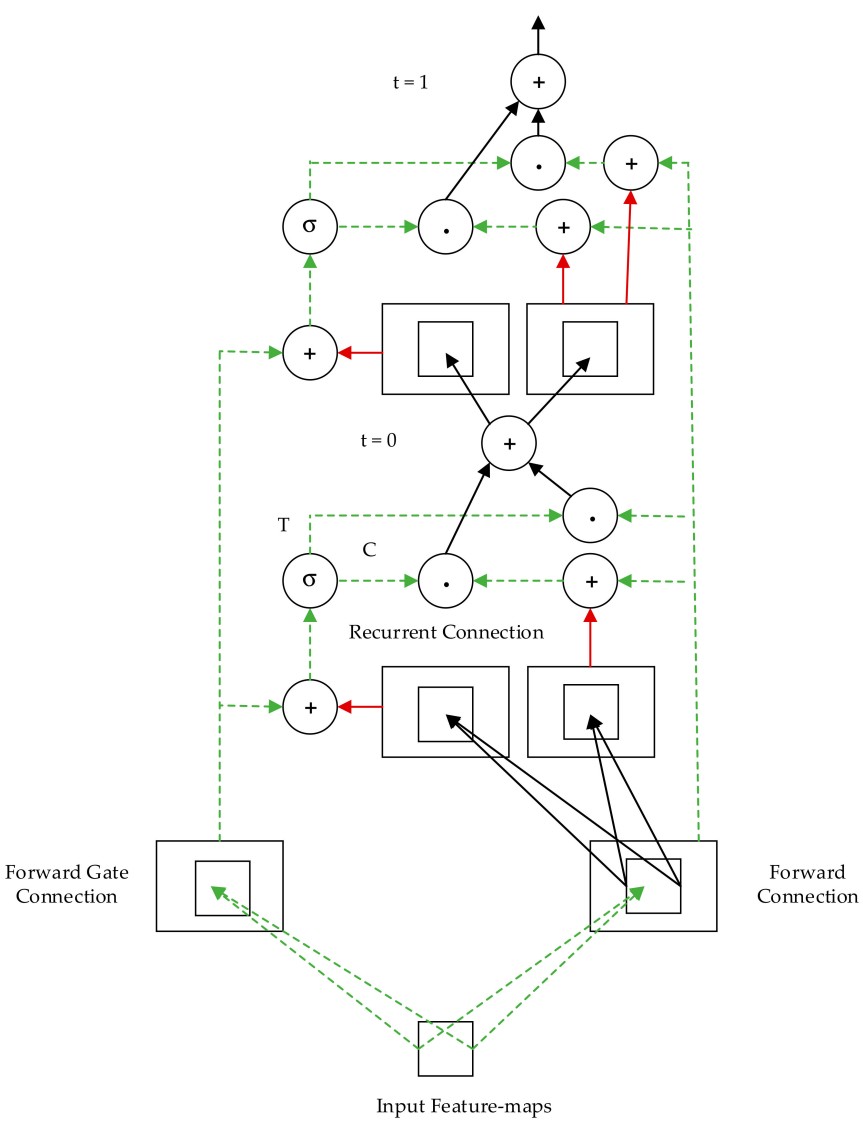

**Figure 4.** Highway recurrent convolutional neural network (HRCL).

Our HRCNN model contains multiple stacked HRCLs and is interleaved with other intermediate layers, such as the fully connected DNN or BGRU and softmax output layers. Each HRCL uses max-pooling to reduce the frequency variations of the feature maps, and the layer normalization, ReLU nonlinearity, and dropout for preventing the overfitting challenge of the model, sequentially, as presented in Table 3.

### 3.5. Res-RCNN and Res-RGCNN Models

The residual network connection is helpful for reducing the computational complexity of the network model by integrating the skip connection within various models [32], for training very deep models [32,34], and for regularizing the models in the SGD training process to reduce the vanishing and exploding gradients and overfitting challenges [32,34]. This network connection does not add extra parameters to the models. However, in the CNN model, if the dimensions of the convolutional layers are variable, this model requires an intermediate transformation convolutional layer with one by one filter size followed by a pooling layer. Hence, at this time, the residual connection adds a minimum number of extra parameters for the CNN model. Moreover, the residual connection is integrated into the network models either per layer or skip per two layers (Figure 5). Several studies have explored different techniques to integrate the residual connection to various network models [33,34,57], verifying that the residual connection is effective for developing optimal acoustic models in terms of performance, computational complexity, and total number of parameters.

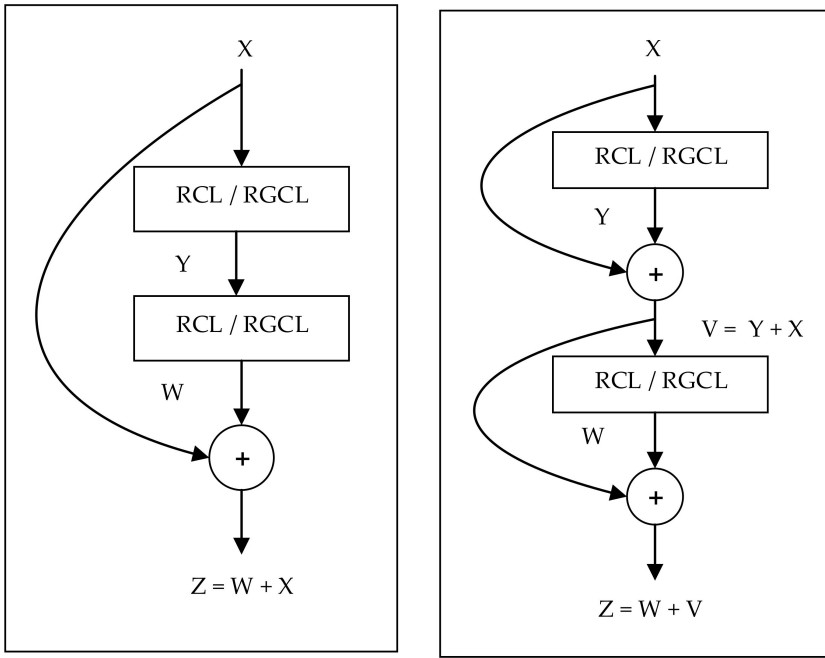

**Figure 5.** The residual connection per two layers (left) and the residual connection per layer (right).

Motivated by the benefits of the residual connection, we propose Res-RCNN and Res-RGCNN models for low-resource speech recognition systems. Our Res-RCNN model contains multiple RCLs stacked together with residual connection per each RCL and interleaved with fully connected DNN or GRU layers and softmax output layer, as shown in Table 2. We use similar operations for each residual RCL as the RCLs of the RCNN model, as described in Section 3.2. Alternatively, integrating the recurrent convolutional connection for the CNN model helps to model the long-term temporal dependencies of feature maps and improve the performance of the CNN model, as discussed in Section 3.2. This enables us to propose the RGCNN model by combining the recurrent convolutional connection with the GCNN model to improve the performance of the model and to increase the context information further. The basic module of our RGCNN model is the recurrent gated convolutional layer (RGCL), and it is defined as:

$$x(t) = h\Big[\big((w^f * u(t)) \odot (w_g^f * u(t))\big) + \big((w^r * x(t-1)) \odot (w_g^r * x(t-1))\big) + b\Big] \tag{12}$$

where $h$, $w^f$, $w^r$, $w_g^f$, $w_g^r$, $*$, $\odot$, and $b$ denote ReLU nonlinearity, forward convolutional weights, recurrent convolutional weights, forward convolutional weights for gate operation, recurrent convolutional

weights for gate operation, convolutional operation, element-wise multiplication operation, and bias of filters, respectively.

Our RGCNN model contains multiple stacked RGCLs and is interleaved with fully connected DNN or BGRU, and the softmax output layers. Each RGCL uses the same operations as the GRCLs. However, the RGCNN model is at great risk of vanishing and exploding gradients challenges, similar to conventional RNN and RCNN models. To overcome these challenges, we propose the Res-RGCNN model by integrating the residual connection with the RGCNN model. The basic component of the proposed model is the residual recurrent gated convolutional layer (Res-RGCL), which is shown in Figure 6. Then, multiple Res-RGCLs are stacked together and interleaved with other intermediate layers, such as fully connected DNN or BGRU layers and softmax output layer, thereby achieving a fully flagged model, called the Res-RGCNN model. Each Res-RGCL of this model is implemented in the same manner as the RGCL of the GRCNN model. Table 3 presents the overall configuration of the Res-RGCNN model.

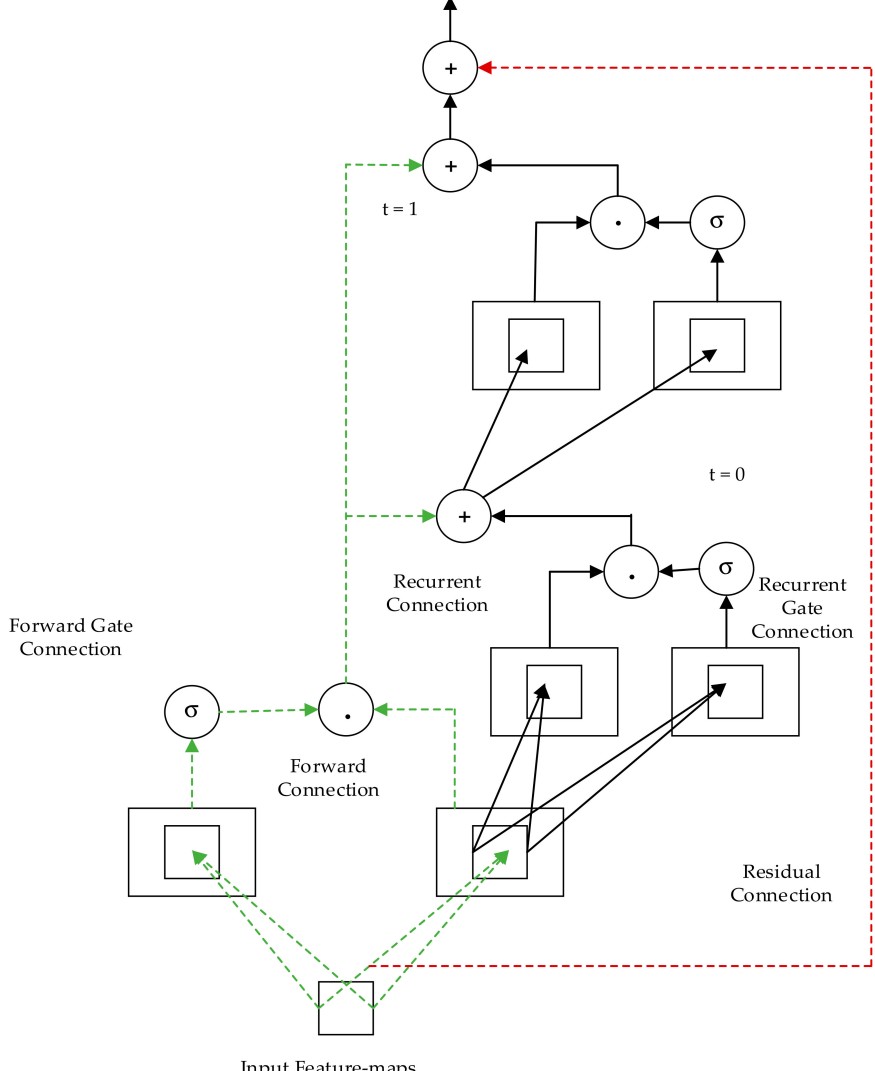

**Figure 6.** Residual recurrent gated convolutional neural network (Res-RGCL).

## 4. Experimental Results

In this section, we describe the effectiveness of our proposed neural network acoustic models, which are described in Section 3, for low-resource-language speech recognition systems. Those models were compared with the baseline neural network acoustic models and with each other via performance,

speed, training time, and model size criteria. This section also describes the preparation of training corpora, and the development of the baseline and proposed models.

### 4.1. Data Corpus

The publicly available corpus was taken from the Babel program. This program is a project that is conducted under the control of IARPA organization to develop various speech recognition and keyword search systems for low-resource languages using a limited amount of training corpus. In this program, a training corpus from 25 languages was collected with certain languages as surprise languages to test the ability of the teams to rapidly build speech recognition and keyword search systems for a new language. The list of the languages and official releases are given in Table 4.

**Table 4.** IARPA Babel program languages and dataset releases.

| Language | ID | Release | Description |
|---|---|---|---|
| Cantonese | 101 | IARPA-babel101-v0.4c | |
| Pashto | 104 | IARPA-babel104b-v0.4aY | Babel option period one languages |
| Tagalog | 106 | IARPA-babel106-v0.2g | |
| Vietnamese | 107 | IARPA-babel107b-v0.7 | |
| Assamese | 102 | IARPA-babel102b-v0.5a | |
| Bengali | 103 | IARPA-babel103b-v0.4b | |
| Haitian Creole | 201 | IARPA-babel201b-v0.2b | Babel option period two languages |
| Lao | 203 | IARPA-babel203b-v3.1a | |
| Tamil | 204 | IARPA-babel204b-v1.1b | |
| Zulu | 206 | IARPA-babel206b-v0.1e | |
| Swahili | 202 | IARPA-babel202b-v1.0d | |
| Kurdish | 205 | IARPA-babel205b-v1.0a | |
| Tok-Pisin | 207 | IARPA-babel207b-v1.0c | |
| Cebuano | 301 | IARPA-babel301b-v2.0b | Babel option period three languages |
| Kazakh | 302 | IARPA-babel302b-v1.0a | |
| Telugu | 303 | IARPA-babel303b-v1.0a | |
| Lithuanian | 304 | IARPA-babel304b-v1.0b | |
| Guarani | 305 | IARPA-babel305b-v1.0b | |
| Igbo | 306 | IARPA-babel306b-v2.0c | |
| Amharic | 307 | IARPA-babel307b-v1.0b | Babel option period four languages |
| Mongolian | 401 | IARPA-babel401b-v2.0b | |
| Javanese | 402 | IARPA-babel402b-v1.0b | |
| Dholuo | 403 | IARPA-babel403b-v1.0b | |

For each of the above language, three different training datasets were prepared: conversational telephone speech, scripted recordings, and far field recordings. These datasets were released with three language packages, such as a full language package of 40-h, a limited language package of 10-h, and a very limited language package of 3-h. Each language also has 10-h development datasets, which are released together with training datasets. There is released lexical dictionary and a language-specific peculiarities document that helps to modify a grapheme-to-phoneme lexical dictionary. Of the 25 Babel languages, the four languages that are released in the third Babel option period (Cebuano (https://catalog.ldc.upenn.edu/LDC2018S07) [58], Kazakh (https://catalog.ldc.upenn.edu/LDC2018S07) [59], Telugu (https://catalog.ldc.upenn.edu/LDC2018S16) [60], and Tok-Pisin (https://catalog.ldc.upenn.edu/LDC2018S02) [61]) were used as our target languages for this study. Hence, the transcribed conversational telephone speech corpus of the limited language package of each language was used as a training corpus. The supplied development dataset was used for evaluating the performance of the models. Moreover, we used two Ethiopic languages, namely Amharic and Chaha, as additional target languages for evaluating our proposed models. Amharic (https://github.com/besacier/ALFFA_PUBLIC/tree/master/ASR/AMHARIC) is a low-resource language,

which has only 26-h read speech training corpus [7]. Chaha (https://m.scirp.org/papers/97733) is a very low-resource language that has only 2.67-h read speech training corpus [42]. This corpus size is increased by generating additional synthetic datasets via the speed perturbed audio augmentation approach. As a result, Chaha has a total of 8.01-h training speech corpus. For our study, we also applied these limited size training corpora of the Amharic and Chaha languages. The description of the training and development datasets of the target languages are given in Table 5.

**Table 5.** Training and development datasets description of the target languages.

| Language | Training Dataset | | | Development Dataset | | |
|---|---|---|---|---|---|---|
| | # of Speaker | # of Sentences | Duration (Hours) | # of Speakers | # of Sentences | Duration (Hours) |
| Amharic | 125 | 13,549 | 26.00 | 20 | 359 | 1.00 |
| Chaha | 15 | 5334 | 8.01 | 5 | 222 | 0.33 |
| Cebuano | 127 | 11,215 | 10.37 | 134 | 11,199 | 10.27 |
| Kazakh | 130 | 11,570 | 9.92 | 140 | 11,678 | 9.73 |
| Telugu | 134 | 11,150 | 10.21 | 120 | 10,120 | 9.74 |
| Tok-Pisin | 127 | 9820 | 9.83 | 132 | 10,156 | 9.92 |

The word-based lexical dictionaries of the Babel languages were prepared by expanding the released lexicons based on the language-specific peculiarities documents from the transcribed training texts, while the word-based lexicons of Amharic and Chaha were prepared using grapheme-based approach by taking the most frequent words from the text corpus of 216,076 sentences [7] and 14,595 sentences [42], respectively. The word-based backoff trigram statistical language models were also developed using the SRILM toolkit [62]. The models of the four Babel languages were trained using 90% of the text transcription of the training speech corpus, while the models of Amharic and Chaha were trained using a text corpus of 216,076 and 14,595 sentences, respectively. All language models were smoothed using the modified Kneser–Ney smoothing algorithm to reduce the word sparseness. The perplexity values of the language models are given in Table 6 on the development test set sentences of Amharic and Chaha languages, as well as on 10% of the text transcription of training speech corpus of the four Babel languages. Table 6 gives a detailed statistical description of the lexicons and language models of the target languages.

**Table 6.** Lexicon size, number of phonemes, and language model perplexity of the target languages.

| Language | Lexicon Size (Number of Words) | Number of Phonemes | Language Model Perplexity |
|---|---|---|---|
| Amharic | 85,000 | 233 | 76 |
| Chaha | 4000 | 41 | 103 |
| Cebuano | 7617 | 28 | 129 |
| Kazakh | 8552 | 61 | 229 |
| Telugu | 15,552 | 50 | 461 |
| Tok-Pisin | 3780 | 37 | 77 |

*4.2. Results of the Baseline Acoustic Models*

The DNN, CNN, conventional BRNN, and BGRU models were developed as baseline models to compare with the proposed advanced neural network acoustic models. Before developing these models, the conventional GMM-HMM system was trained to obtain the target senones using the Kaldi toolkit [63]. The three-state left-to-right HMM topology with the fourth-last non-emitting state was used for building several phoneme-based speech recognition systems for Chaha, Cebuano, Kazakh, Telugu, and Tok-Pisin languages. The five-state Bakis HMM topology with the sixth-last non-emitting state was also applied for developing various syllable-based speech recognition systems for Amharic

language. The GMM-HMM systems were trained using 13-dimensional Perceptual Linear Prediction (PLP) features concatenated with 3-dimensional pitch features with zero means and unit variances. Then, LDA and MLLT were applied to make the features more accurately modeled by the diagonal covariance Gaussians to reduce the feature dimensionality to 40. Next, GMM-HMM was trained with SAT using fMLLR and further enhanced by discriminative training using the boosted maximum mutual information (BMMI) criterion. Consequently, the number of senones obtained from the GMM-HMM systems are 3948, 960, 1968, 1984, 1904, and 1968 for Amharic, Chaha, Cebuano, Kazakh, Telugu, and Tok-Pisin languages, respectively.

The DNN model contains five hidden layers with 1024 dimensions. The input features are 40-dimensional Mel-scale Filterbank concatenated with three-dimensional pitch features with first- and second-order derivatives. These features were normalized to zero means and unit variances. The left and right context widths of five were used to combine the frames. ReLU was used as an activation function. A dropout rate of 0.15 and a batch size of 128 were utilized for training the model for 15 epochs using the SGD algorithm with cross-entropy loose function. The initial learning rate of 0.08 was used, which was reduced by a factor of two when the model improvement thresholding was less than 0.001. The batch normalization technique was also applied for preventing the model overfitting challenge.

The CNN model consists of two convolutional hidden layers with 80 filters and 3 filter lengths, with padding value of 1; one max-pooling hidden layer with a pooling size of 3 between the convolutional layers; and four fully connected hidden layers with 1024 nodes. The input features for CNN are the same as those for DNN, explicitly 40-dimensional Mel-scale Filterbank concatenated with 3-dimensional pitch features with first- and second-order derivatives. A context width of 11 frames with five left and five right frames was used. Therefore, the input feature map was divided into 43 bands with 33 feature maps per band. The initial learning rate of 0.08 per frame and a batch size of 128 were applied. The learning rate was managed similarly to DNN. The CNN was trained for 15 epochs. Layer-normalization and batch-normalization were used for convolutional and fully connected hidden layers, respectively. A dropout rate of 0.15 was applied for all convolutional and fully connected layers.

The conventional BRNN model consists of three hidden layers with 550 hidden states. The input features are similar to the DNN models, which are 40-dimensional Mel-scale Filterbank with 3-dimensional pitch features with first- and second-order derivatives, and normalized to zero means and variances. The dropout rate of 0.2 and batch normalization were used to monitor the model overfitting challenge. The model was trained using a batch of 16 for 15 epochs using the RMSprop optimization algorithm with an initial learning rate of 0.00032, and it was halved when the model improvement threshold was less than 0.001. We used ReLU nonlinearity as an activation function for hidden layers. The optimization parameters used are alpha with a value of 0.95 and eps with the value of $1e^{-8}$. At the time of training, the sequence length was increased by half starting from the initial sequence length of 100 up to the maximum sequence length of 500. On the other hand, the BGRU model used all the parameter values of the conventional BRNN model except the initial learning rate and the hidden layer activation function. This model used an initial learning rate of 0.00004 and tanh nonlinearity as an activation function for hidden layers.

The entire baseline and proposed neural network models were trained using the Pytorch-Kaldi [64] toolkit. This toolkit uses the PyTorch deep learning framework for implementing the neural network models and Kaldi for feature extraction and decoding. Therefore, our proposed models were implemented using the PyTorch deep learning framework and run on the PyTorch-Kaldi toolkit. The training process was accelerated using Nvidia Tesla M40 GPU on a single machine from Alibaba cloud computing services. A Kaldi weighted finite-state transducer decoder with a beam size of 13.0, a lattice beam size of 4.0, and an acoustic weight of 0.1 was applied for all experiments. The WER evaluation metric was used for all target languages. The results of the baseline models are given

in Table 7. These results are comparable with some of the previous published studies on the target languages [7,34,42,43,47,48,65].

**Table 7.** WER (%) of the baseline neural network models for all target languages.

| Model | WER (%) | | | | | |
|-------|---------|-------|---------|--------|--------|----------|
|       | Amharic | Chaha | Cebuano | Kazakh | Telugu | Tok-Pisin |
| DNN   | 11.35   | 23.70 | 68.33   | 70.75  | 86.20  | 50.14    |
| CNN   | 10.30   | 22.75 | 67.23   | 70.08  | 86.37  | 49.35    |
| BRNN  | 12.76   | 25.82 | 72.63   | 74.85  | 89.70  | 54.71    |
| BGRU  | 9.89    | 22.36 | 66.41   | 69.38  | 84.98  | 48.22    |

### 4.3. Results of the Proposed Neural Network Models

This section presents the empirical results of the proposed advanced neural network acoustic models for low-resource speech recognition. During the training of these models, the major hyperparameters were tuned using the Tok-Pisin language. Then, the optimal models were trained for the rest target languages.

#### 4.3.1. Gated Convolutional Neural Network Model

The network configuration for training the GCNN model is given as follows: the input features of the GCNN model were the same as with the CNN model, which is a 40-dimensional Mel-scale Filterbank concatenated with 3-dimensional pitch features with first- and second-order derivatives. The context width of 11 frames (five left and five right contexts) was also applied. The model consists of three GCLs, a max-pooling layer with pooling sizes of 3 and 2 for the first and the second GCLs, and four fully connected hidden layers. An initial learning rate of 0.08 per frame and a batch size of 128 were used for training the GCNN model. The learning rate was managed as described for the CNN model. The model was trained for 15 epochs. Layer-normalization and batch-normalization were applied for gated convolutional and fully connected hidden layers, respectively. A dropout regularizer with a rate of 0.15 was utilized for all gated convolutional and fully connected hidden layers.

The filter sizes and filter lengths of the GCLs are the two major hyperparameters of the GCNN model. Those hyperparameters were tuned to acquire the optimal values for the model. First, the filter sizes of GCLs were tuned by making the filter length fixed to 3. A filter size of 60 was optimal for the first, second, and third GCLs (Figure 7). Second, the filter length of the GCLs was tuned using the optimal values of filter sizes that were obtained previously, and a filter length of 3 was optimal for training the GCNN model (Figure 8).

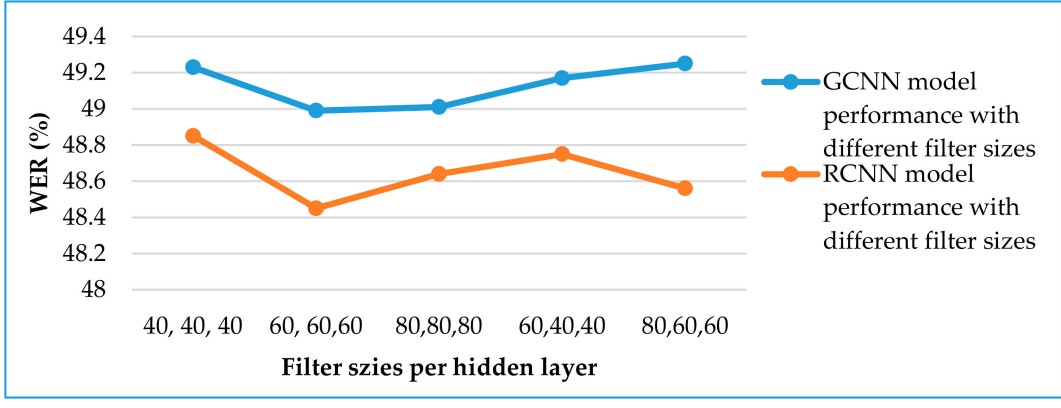

**Figure 7.** WER results vs. filter sizes of hidden layers for the GCNN and RCNN models.

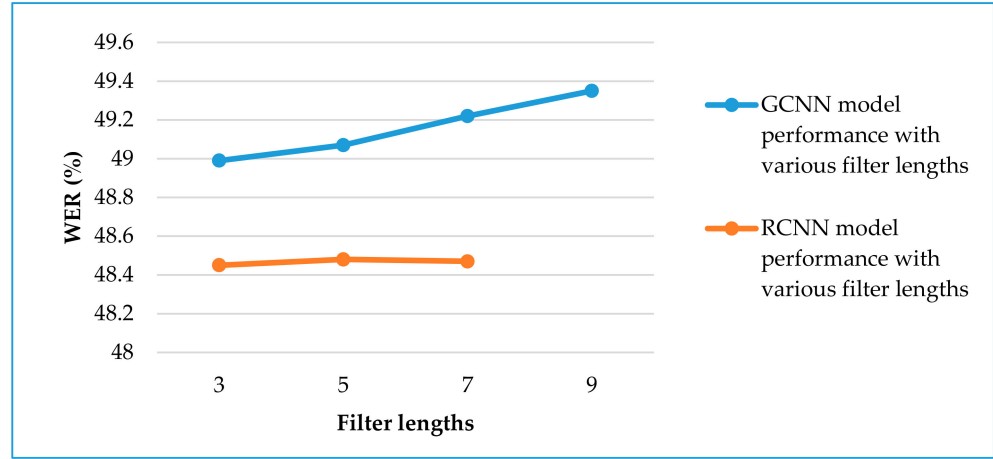

**Figure 8.** WER results vs. the filter lengths for the GCNN and RCNN models.

The GCNN model achieves good results for the Tok-Pisin language. To assure the quality of the model performance, we conducted experiments using the best GCNN model for all target languages. The experimental results are presented in the fifth row of Table 8. The results show that the GCNN model outperforms the baseline conventional DNN, CNN, and BRNN models with relative performance improvements of 0.21–12.33%, 0.1–3.4%, and 4.1–22.02%, respectively, for all target languages. This is because the gate connections can model longer frame context. However, this model is worse than the BGRU model, with a relative WER of 0.1–1.57% for all target languages.

**Table 8.** WER (%) of the baseline and proposed neural network models for all target languages (best performed acoustic models are shown in bold).

| Remark | Model | WER (%) | | | | | |
|---|---|---|---|---|---|---|---|
| | | Amharic | Chaha | Cebuano | Kazakh | Telugu | Tok-Pisin |
| **Baseline neural network models** | DNN | 11.35 | 23.70 | 68.33 | 70.75 | 86.20 | 50.14 |
| | CNN | 10.30 | 22.75 | 67.23 | 70.08 | 86.37 | 49.35 |
| | BRNN | 12.76 | 25.82 | 72.63 | 74.85 | 89.7 | 54.71 |
| | BGRU | 9.89 | 22.36 | 66.41 | 69.38 | 84.98 | 48.22 |
| **Proposed neural network models** | GCNN | 9.95 | 22.35 | 67.08 | 70.01 | 86.02 | 48.99 |
| | RCNN | **9.63** | **22.05** | 66.66 | 69.27 | 85.62 | **48.45** |
| | GRCNN | 9.73 | 22.11 | **66.63** | 69.29 | **85.40** | 48.49 |
| | HRCNN | 9.83 | 22.19 | **66.58** | **69.25** | **85.33** | 48.58 |
| | Res-RCNN | **9.66** | **22.02** | 66.64 | 69.27 | 85.55 | 48.44 |
| | Res-RGCNN | **9.58** | 21.93 | 66.49 | **69.17** | 85.25 | **48.37** |
| | CNN-BGRU | 9.35 | 21.69 | 66.33 | 69.28 | 84.94 | 48.15 |
| | GCNN-BGRU | 8.85 | 21.14 | 66.03 | 69.08 | 84.18 | 47.39 |
| | RCNN-BGRU | 8.20 | 20.38 | 65.35 | 67.57 | 83.33 | 46.25 |
| | GRCNN-BGRU | 8.00 | 20.15 | **65.26** | 67.53 | **82.77** | **46.23** |
| | HRCNN-BGRU | 7.85 | 19.97 | 65.11 | **67.39** | 82.85 | 46.18 |
| | Residual RCNN-BGRU | 7.92 | 20.05 | 65.33 | 67.51 | 83.30 | 46.23 |
| | Residual RGCNN-BGRU | **7.30** | **19.78** | **65.00** | **67.25** | 82.90 | **45.95** |
| Best case absolute WER (%) reduction | | 5.46 | 6.04 | 7.63 | 7.6 | 6.8 | 8.76 |

### 4.3.2. Recurrent Convolutional Neural Network Model

The RCNN model was trained using 40-dimensional Mel-scale Filterbank features concatenated with 3-dimensional pitch features with the first and second derivatives. A context width of 11 frames (five left and five right contexts) was applied. A dropout regularizer with a rate of 0.15 was used for both the recurrent convolutional and fully connected hidden layers. We used three recurrent convolutional and four fully connected hidden layers. An initial learning rate of 0.08 was applied, and managed as described for the CNN model. The training algorithm is similar to the conventional

CNN model and the RCNN model is also trained for 15 epochs. All other model parameters were similar to the baseline conventional CNN model.

The filter sizes, filter lengths, and recurrent time-steps are the major hyperparameters of the RCNN model. These parameters were tuned to obtain the optimal values for the model. Different filter sizes of RCLs were tested by using a filter length with the value of 3 and time steps with the value of 2. The results indicate that the filter size of 60 was optimal for the first, second, and third RCLs (Figure 7). Various filter lengths of the RCLs were examined using the optimal filter size, which was obtained before, and by making the time steps value to 2. Figure 8 presents the performance of the RCNN model with variable filter lengths. Accordingly, the filter length of three is an optimal value. The time-steps for the recurrent connection were also investigated using the optimal values of the filter size and filter length (Table 9). The results show that the optimal value of recurrent time-step is 3.

**Table 9.** RCNN model performance with various time-steps of the recurrent connections (best value is shown in bold).

| Time-Steps | WER (%) |
|:----------:|:-------:|
| 1 | 48.46 |
| 2 | **48.45** |
| 3 | 48.48 |

The RCNN model provides better results for the Tok-Pisin language. To ensure the effectiveness of this model for the remain target languages, we conducted further experiments using the best RCNN model. The experiential results are presented in the sixth row of Table 8. The results show that the RCNN model boosts the baseline DNN, CNN, and BRNN models with relative performance improvements of 0.67–15.15%, 0.85–6.5%, and 4.55–24.53%, respectively, for all target languages. The RCNN model outperforms the corresponding BGRU model with a relative performance improvement of 2.63%, 1.39%, and 0.16% for the Amharic, Chaha, and Kazakh languages, respectively, while it is worse than that of the BGRU model, with relative WER of 0.38–0.75% for the remaining languages. The RCNN model also outperforms the GCNN model with performance improvements of 0.63–3.22% for all target languages. This is because the recurrent connections are able to model long-term context dependencies among feature frames and the effectiveness of CNN model for low-resource languages.

### 4.3.3. Gated Recurrent Convolutional Neural Network Model

The network configuration of the GRCNN is described as follows. The input features for the GRCNN model were the same as those of the RCNN model: 40-dimensional Mel-scale Filterbank concatenated with 3-dimensional pitch features with first- and second-order derivatives. The context width of 11 frames was applied. Hence, the input feature map was divided into 43 bands with 33 feature maps per band. The filter length of 3 and padding of 1 were used for all GRCLs. We used max-pooling layers with pooling size of 3 and 2 over the top of first and second GRCLs, and no max-pooling layer on the top of last GRCL. The four fully connected hidden layers with 1024 dimensions were added on the top of last GRCL. An initial learning rate of 0.08 per frame and a batch size of 128 were used. The learning rate was managed as described for the RCNN model. GRCNN was trained for 15 epochs. Layer-normalization and batch-normalization were applied for GRCLs and fully connected hidden layers. A dropout rate of 0.15 was applied for all GRCL and fully connected hidden layers. Weight-decay was also used with a value of 0.0001 for preventing a model from overfitting challenge. The hyperparameters of the GRCNN model includes the number of GRCLs and the GRCL dimensions. Various numbers of GRCLs were tuned by making the GRCLs dimensions fixed to 80 (Figure 9). The results show that the optimal number of GRCLs is 3 for training the GRCNN model. Similarly, the GRCL dimensions were tuned by setting the number of GRCLs to 3, and the results are presented in Figure 10. The results demonstrate that the optimal number of dimensions for the GRCL is 80.

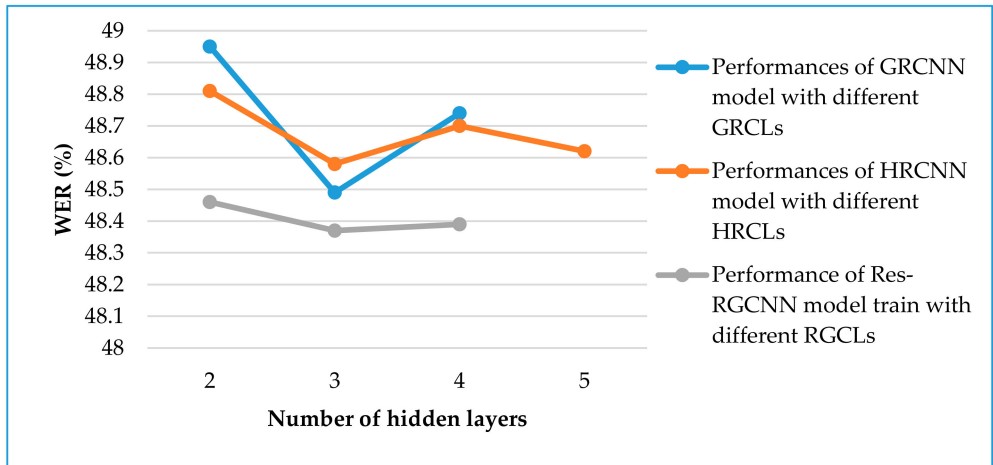

**Figure 9.** WER results vs. the number of hidden layers for the GRCNN, HRCNN, and Res-RGCNN models.

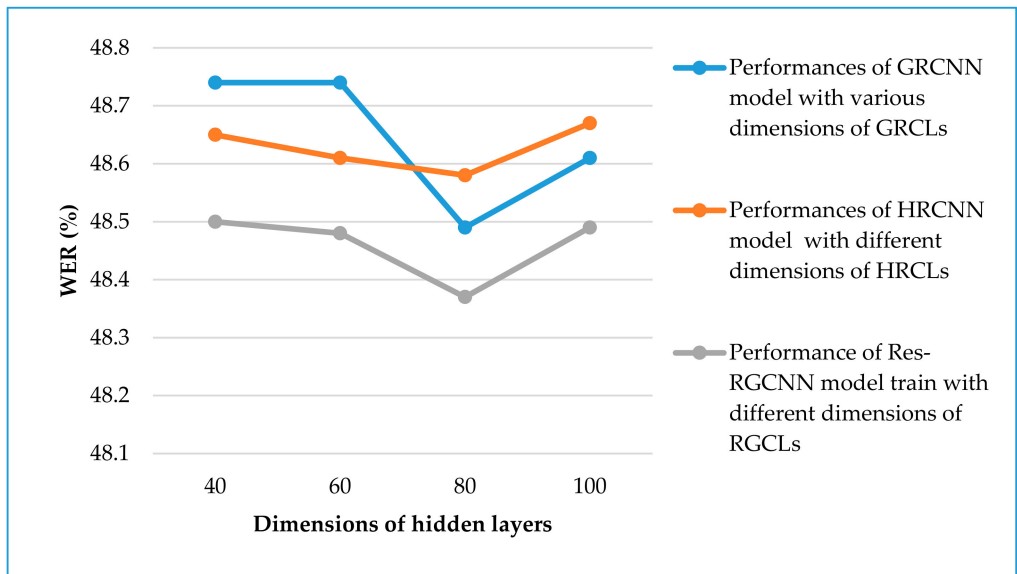

**Figure 10.** WER results vs. dimensions of hidden layers for the GRCNN, HRCNN, and Res-RGCNN models.

Using the best GRCNN model trained for the Tok-Pisin language, we trained the GRCNN model for all target languages to evaluate the quality of the model. The results are presented in the seventh row of Table 8, and demonstrate that the GRCNN model is better than the baseline models with a relative WER reduction of 0.89–23.75%. This approach also outperforms the GCNN and RCNN models with relative performance improvements of 0.67–2.21% and 0.05–0.26%, respectively, for all target languages. This is because the gate connection models long-term frame context dependencies and helps to reduce the gradient vanishing problems across several GRCLs through controlling the flow of feed-forward and recurrent information.

### 4.3.4. Highway Recurrent Convolutional Neural Network Model

The model parameters of the HRCNN model include input features of 40-dimensional Mel-scale Filterbank features concatenated with 3-dimensional pitch features, with the first and second derivatives used as in the previous models. Those features were combined using the context width of 11 frames, containing five left and five right frames. The training algorithm was similar to that of the GRCNN models. For tuning the optimal number of HRCLs, we used filter sizes of 80. Next, using the obtained optimal number of HRCLs, we tuned the optimal number of HRCL dimensions. Figures 9 and 10 show the performances of the HRCNN model with different numbers of HRCLs and dimensions of HRCL,

respectively. The results show the optimal values of the HRCLs, and the dimensions of the HRCL were 3 and 80, respectively.

The HRCNN model gives better performance for the Tok-Pisin language over the baseline models. To evaluate the model effectiveness for other languages, we trained the HRCNN model for all target languages, and the results are presented in the eighth row of Table 8. The results demonstrate that the HRCNN model outperforms the baseline models with WER reductions of 0.97–22.96% for all target languages. The HRCNN model also gives a comparable performance with our GRCNN and RCNN models. This is because the highway connections can model the long-term frame context dependencies and learn additional information from the original and intermediate speech signals, as well as avoid the gradient vanishing problems across multiple HRCLs.

### 4.3.5. Residual Recurrent Convolutional Neural Network Model

The Res-RCNN model used the same model parameters as the RCNN model. It contains the residual connection per each RCL, which does not have additional parameters over the RCNN model. All the model configurations of the RCNN model were used for this model and trained for all target languages. The performance of the Res-RCNN model is presented in the ninth row of Table 8. The results show that the Res-RCNN model reduces the WER of the baseline models by 0.75–24.29% and outperforms the RCNN model with performance enhancements of 0.02–0.14% for target languages. This is because the residual connection learns more information from the original and intermediate speech signals, and helps to avoid the gradient vanishing problems across multiple Res-RCLs.

### 4.3.6. Residual Recurrent Gate Convolutional Neural Network Model

The Res-RGCNN model was trained using the same input features as in the previous models, which is 40-dimensional Mel-scale Filterbank features concatenated with 3-dimensional pitch features and with the first and second derivatives. A total context width of 11 frames was applied, and all other model parameters were similar to those of the GCNN model. The number of residual RGCLs and their corresponding dimensions are the major hyperparameters of the Res-RGCNN model. These parameters were tuned to achieve optimal values. Accordingly, the number of residual RGCLs was tuned by making the dimensions of residual RGCLs fixed to 80 nodes (Figure 9). The results show that the optimal values of RGCLs are 3. Likewise, the dimensions of the residual RGCLs are tuned using the optimal number of residual RGCLs (Figure 10). The results demonstrate that the optimal value of the dimensions of the residual RGCLs is 80 nodes.

We found that the Res-RGCNN model is effective for the Tok-Pisin language. However, to evaluate the effectiveness of this model further, we trained the best Res-RGCNN model for the remain target languages, and the results are presented in the tenth row of Table 8. These results demonstrate that the Res-RGCNN model is the best performing model of all the baseline and proposed neural network models, providing performance improvements of 1.10–24.92% and 0.1–3.72% over the baseline and our previously proposed models, respectively. This is because the residual connection learns more information from the original and intermediate speech signals, and helps to avoid the gradient vanishing problems across multiple Res-RGCLs.

### 4.3.7. Various CNN-BGRU Neural Network Models for All Languages

Most conventional CNN-based neural network models use the fully connected DNN layers on the top of the last convolutional or pooling layer. Because of this, all our proposed advanced neural network models use the fully connected DNN layers on the top of the model-specific layer or pooling layer. In this test, the fully connected neural network layers were replaced with BGRU layers for modeling the long-term context dependencies of the features. Accordingly, the fully connected neural network layers of the classical CNN, GCNN, RCNN, GRCNN, HRCNN, Res-RCNN, and Residual RGCNN neural network models were replaced by three BGRU layers, and trained for all target languages. The network configuration of the added BGRU layers was similar to the baseline BGRU

layers for all neural network models. The experimental results are presented in Rows 11–17 of Table 8. The results show that the various CNN-BGRU neural network models outperform the corresponding baseline and fully connected neural network layers-based CNN models, with relative performance improvements of 0.5–42.79% and 1.33–23.8%, respectively. The best performing acoustic models are GRCNN-BGRU, HRCNN-BGRU, and Res-RGCNN-BGRU with absolute WER reductions of 1.15–8.48, 1.99–8.53, and 1.41–8.76. These results demonstrate the effectiveness of integrating the BGRUs to one-dimensional frequency CNN models with the temporal context.

### 4.3.8. Training Time, Recognition Speed, and Parameters of the Models

The training time and recognition speed of neural network models depend on the computational capacity of the machine. Accordingly, all neural network models were trained using a single NVIDIA Tesla M40 GPU. The training time and recognition speed of all the models for the target languages are presented in Table 10. The results demonstrate that the training times per epoch of the DNN model are 2.8-, 2.0-, 5.4-, 11.1-, 11.9-, 5.1-, and 12.1-times faster than the conventional CNN, GCNN, RCNN, GRCNN, HRCNN, Res-RCNN, and Res-RGCNN models, respectively. Of the proposed acoustic models, the GRCNN, HRCNN, Res-RGCNN, RCNN-BGRU, GRCNN-BGRU, HRCNN-BGRU, Res-RCNN-BGRU, and Res-RGCNN-BGRU models are much more expensive and slower models than the other neural network models. Moreover, the training time of the BGRU model is about 4.5-times slower than the DNN model, and it is more efficient than all the convolutional recurrent models.

**Table 10.** Training time per epoch (hours), recognition speed (RTF), and model size (millions) of the baseline and proposed neural network models for all target languages (best performed acoustic models are shown in bold).

| Remark | Model | Training Time per Epoch (Hours) | | | | | | Average RTF per Model | Model Size |
|---|---|---|---|---|---|---|---|---|---|
| | | Am. | Ch. | Ceb. | Ka. | Tel. | Tok. | | |
| **Baseline neural network models** | DNN | 0.25 | 0.09 | 0.11 | 0.15 | 0.17 | 0.16 | 9.788 | 7.7 |
| | CNN | 0.70 | 0.33 | 0.37 | 0.35 | 0.39 | 0.38 | 7.651 | 13.9 |
| | BRNN | 0.38 | 0.17 | 0.20 | 0.19 | 0.22 | 0.21 | 11.814 | 2.2 |
| | BGRU | 1.12 | 0.54 | 0.61 | 0.56 | 0.63 | 0.59 | 9.756 | 8.7 |
| **Proposed neural network models** | GCNN | 0.5 | **0.23** | **0.25** | **0.27** | **0.29** | **0.28** | **8.113** | 19.8 |
| | RCNN | **1.40** | **0.66** | **0.69** | **0.70** | **0.73** | **0.71** | **7.865** | 19.8 |
| | GRCNN | 2.90 | 1.30 | 1.45 | 1.46 | 1.48 | 1.45 | 8.143 | 24.8 |
| | HRCNN | 3.04 | 1.50 | 1.53 | 1.55 | 1.54 | 1.52 | 8.8599 | 24.8 |
| | Res-RCNN | **1.38** | **0.67** | **0.69** | **0.73** | **0.70** | **0.71** | **7.865** | 19.8 |
| | Res-RGCNN | 3.08 | 1.52 | 1.55 | 1.56 | 1.58 | 1.54 | 8.8745 | 25.9 |
| | CNN-BGRU | 1.82 | 0.88 | 0.98 | 0.91 | 1.02 | 0.97 | 7.764 | 23.9 |
| | GCNN-BGRU | **1.73** | **0.82** | **0.86** | **0.87** | **0.92** | **0.87** | **8.342** | 28 |
| | RCNN-BGRU | **2.55** | **1.22** | **1.30** | **1.26** | **1.36** | **1.30** | **7.849** | 28 |
| | GRCNN-BGRU | 4.08 | 2.00 | 2.14 | 2.10 | 2.11 | 2.04 | 8.523 | 34.5 |
| | HRCNN-BGRU | **2.22** | 2.08 | 2.14 | 2.11 | 2.17 | 2.11 | 8.872 | 34.6 |
| | Residual RCNN-BGRU | 2.60 | **1.27** | **1.30** | **1.29** | **1.33** | **1.30** | **7.912** | 28 |
| | Residual RGCNN-BGRU | 4.40 | 2.09 | 2.16 | 2.12 | 2.21 | 2.13 | 8.976 | 36.6 |

Keyword: Am, Amharic; Ch, Chaha; Ceb, Cebuano; Ka, Kazakh; Tel, Telugu; Tok, Tok-Pisin.

Table 10 shows the recognition speeds of all the developed acoustic models. The results demonstrate that the proposed models are 8.0–33.6% faster than the baseline DNN, BRNN, and BGRU models, while the baseline CNN model is 2.5–14.8% faster than the proposed models. Overall, the recognition speeds of the entire acoustic models are relatively slow. Table 10 also demonstrates that the proposed models have 1.4–16.6 times more parameters than the baseline models. This is because the proposed models are the combination of several neural network acoustic models.

Figure 11 illustrates the frame accuracies for various models on the development set for Tok-Pisin. The figure shows that all the proposed models have better development set accuracy than the baseline DNN, CNN, and BRNN models. This is because the recurrent connections, network connections (gate, highway, and residual), and model regularization (e.g., dropout, layer, and batch normalizations)

improve the overall generalization abilities of the models. The figure also shows that the proposed models have a quick convergence on the development set than the baseline models.

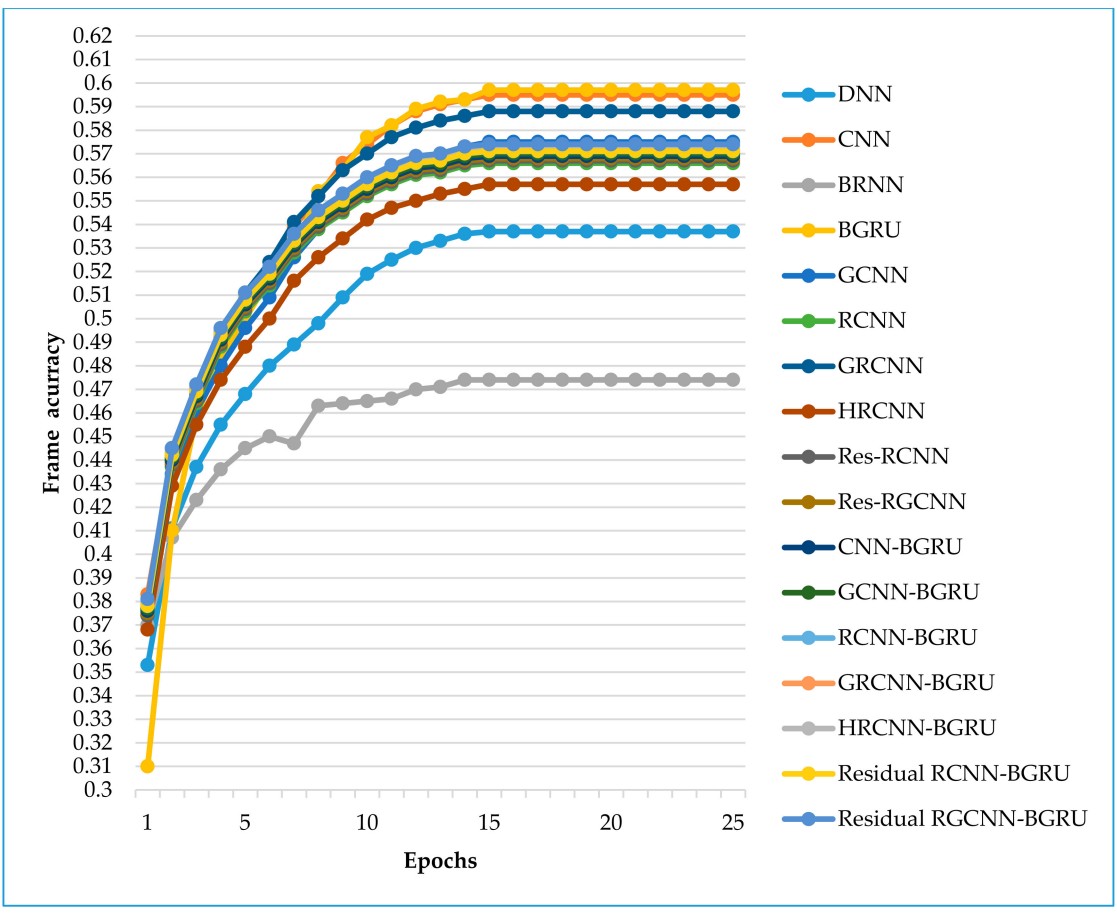

**Figure 11.** The frame accuracy for different models on the development set for Tok-Pisin.

## 5. Discussion

Various baseline and proposed neural network models were tested to build acoustic models for speech recognition systems of low-resource languages. The evaluation metrics, including WER to measure the accuracy of the models, training time to measure the time taken to train the models, real time factor (RTF) to measure the delay of the recognition speed compared with the user speeches, and model size are the total parameters of the model. The overall performances of the various neural network models developed for all target languages are presented in Table 8. The results demonstrate that the proposed neural network acoustic models (GCNN, RCNN, GRCNN, HRCNN, Res-RCNN, and Res-RGCNN) outperform the baseline neural network acoustic models (e.g., DNN, CNN, BRNN, and BGRU). Besides, our proposed acoustic models are superior in performance relative to the DNN [7,34,47,48], time-delay neural network (TDNN) [42,43], CNN [34], and very deep CNN [34] models, which are presented in Section 2. This is because of the capabilities of the gate, highway, and residual connections integrated to CNN and RCNN models. The gate connection adds additional parameters to the modes. It can model the long-term frame context dependencies and prevents model overfitting and gradient vanishing and exploding challenges, which are often encountered during the training process. Similarly, the highway and residual connections are also better connections to model the long-term frame context dependencies and learn additional information from the original and intermediate speech signals, as well as avoid the gradient vanishing and exploding problems across multiple hidden layers of our proposed acoustic models. Of the proposed models, Res-RGCNN, Res-RCNN, and HRCNN are the best performing models for all target languages.

On the other hand, when the fully connected hidden layers of the proposed models are replaced by the BGRU layers, as described in Section 4.3.7, all CNN-BGRU models outperform the corresponding baseline and fully connected hidden layers-based proposed models for all target languages. This shows that the one-dimensional CNN model is compatible with the BGRU layers for modeling the speech sequence, similar to how the fully connected hidden layers are compatible with the two-dimensional CNN models. The Res-RGCNN-BGRU, HRCNN-BGRU, and GRCNN-BGRU models are the best performing neural network models among the various other CNN-BGRU models.

The proposed acoustic models were also compared with the baseline acoustic models using the training time, recognition speed, and total number of parameters. Based on the training time, both the fully connected layers-based and BGRU layers-based advanced acoustic models are expensive and slower than the baseline neural network acoustic models. This is because the local weight connections in all the convolutional structures require many small matrix multiplications, which are less efficient for GPU. Moreover, the recurrent connection of the proposed models increases the model computation time. Based on the recognition speed, the proposed advanced acoustic models have faster recognition speed in terms of real time factor than the baseline acoustic models. However, the overall recognition speed of all models are slow. This is because the PyTorch-Kaldi is a hybrid toolkit, with relatively long decoding and loading times via the Kaldi toolkit. The recognition speed also depends on the language model graph size and the size of the testing dataset. Therefore, the decoding speed of all the neural network models varies for the target languages. Based on total number of parameters, the proposed models have more parameters compared with the baseline models. This is because the proposed acoustic models are advanced that are developed through the integration of the baseline (CNN and RNN) models and various connection networks, such as gate, recurrent, highway, and residuals that increase the total number of model parameters.

Generally, even if the proposed advanced CNN acoustic models are worse in terms of training time and total number of parameters than the baseline acoustic models, these models give better recognition performance and fast recognition speed over the baseline models in speech recognition of low-resource languages. Therefore, all the proposed advanced CNN acoustic models are effective to develop low-resource-languages speech recognition systems.

## 6. Conclusions and Future Directions

In this paper, we explore various advanced convolutional neural network acoustic models through the integration of CNN, conventional RNN, and neural network connections (gate, highway, and residual) for low-resource-languages speech recognition systems, aimed to reduce the overfitting limitations of RNN models and to model long-term context feature dependencies using the CNN model. These models include GCNN, RCNN, GRCNN, HRCNN, Res-RCNN, and Res-RGCNN. All those models can also be combined with BGRU layers by replacing the fully-connected hidden layers. The performances of those advanced neural network models were evaluated using two Ethiopic languages, as well as four languages from the IARPA Babel datasets. Our results demonstrate that all the proposed models decrease the WER (by about 0.21–15.59%, 0.1–6.99%, and 4.10–24.92%) relative to the baseline DNN, CNN, and BRNN across the selected six target languages. Moreover, the combination of the proposed and BGRU models decrease the WER by about 0.5–42.79% relative to the baseline DNN, CNN, BRNN, or BGRU models, and by about 1.33–23.8% relative to the proposed GCNN, RCNN, GRCNN, HRCNN, Res-RCNN, and Res-RGCNN models, which contain the fully connected hidden layers. Overall, all the proposed advanced neural network acoustic models are effective for low-resource-languages speech recognition systems. In this study, we used limited language packages of the IARPA Babel datasets. As future work, we are interested in exploring the effectiveness of our proposed neural network models for the full language packages of the IARPA Babel datasets and other low-resource-languages speech recognition systems. We will also explore new neural network acoustic models that integrate the CNN model with the advanced RNN models, such as LSTM and GRU, and evaluate for speech recognition of low-resource languages.

**Author Contributions:** Conceptualization, T.G.F. and T.T.H.; Investigation, J.Y.; Methodology, T.G.F.; Project administration, J.Y.; Resources, T.T.H.; and Supervision, J.Y. All authors have read and agreed to the published version of the manuscript.

**Funding:** This project is fully supported by Chinese Government Scholarship.

**Conflicts of Interest:** The authors declare no conflict of interest.

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
