# Peer review of "Advanced Convolutional Neural Network-Based Hybrid Acoustic Models for Low-Resource Speech Recognition"

_computers, doi:10.3390/computers9020036_

Round 1

Reviewer 1 Report

The paper deals with the problem of low-resource speech recognition and proposes several neural network models that combine convolutional neural network (CNN) and recurrent neural networks (RNNs) with gate, highway, and residual networks.

The paper deals with actual topic with innovative potential in scope of contemporary research in this area. The proposed models and the performed analysis comply with a high technical standard and are described in sufficient detail. The paper is altogether well-written; the proposed research is interesting and well described and the performed analysis provides interesting results, even if the presentation should be improved according to the following suggestions.

Section 2

The baseline acoustic models are quite clear, however there are several imprecisions that should be checked and corrected, such as the following I have noticed:

-in equation 3 the term In should be defined.

-line 145: I think you mean modelling instead of modeming.

-the sentence 'Recurrent neural networks (RNNs) are a neural network approaches' should be corrected

-lines 158 and 179: the sentence 'It represented by' should be corrected in 'It Is represented by'

-Line 168: those should be changed to these.

-Line 195:an empty space should be left after a and b

-Lines 205-207: the sentence should be rewritten as it Is not well formulated.

-Line 194: yk instead of y0 should be defined.

-Equations 12 and 13: the conditions t=0 and t>0 should be spaced and preceded by if.

-Line 253: shard should be corrected in shared.

Section 3

The proposed neural network approaches for speech recognition of low-resource languages are very interesting and provides an enhancement respect to the current knowledge.

Line 390: detail should be corrected in detailed.

Author Response

Dear Reviewer

Thanks for your time and comments on our paper. I have uploaded a file on which I reply to each and every comment provided by you. 

Regards

Reviewer 2 Report

Dear author(s),

thank you for this interesting work. I think with some improvement it can definitely be accepted to be published in this top journal:

Introduction should focus on the novelty in comparison with other works in the area and end up with a clear problem to be solved in the article.

State of the art review should be a sepparte section and it should be more structured and grouped by the methods overviewed. I suggest adding a summarizing table at the end of this section, where all methods would be compared, focusing on less common languages (speech recognizer development wise). I help with less popular languages (speech engine wise) in EU:

  • Polish: https://ieeexplore.ieee.org/abstract/document/8166885/
  • Croation: https://link.springer.com/article/10.1007/s11517-019-01963-6
  • Lithuanian https://doi.org/10.3390/computers8040076
  • Estonian: https://arxiv.org/abs/1901.03601
  • Serbian: http://www.mathnet.ru/php/archive.phtml?wshow=paper&jrnid=trspy&paperid=1006&option_lang=eng
  • Finnish: https://aaltodoc.aalto.fi/handle/123456789/30638
  • Romanian: https://ieeexplore.ieee.org/abstract/document/7990443

Section on neural networks contains a lot of generic and trivial information. I suggest removing it (dropping section two completely) and replacing it by added links in section 3, focusing only on what have you modified yourself.

Please supplement section 3 with all configuration parameters in table format. Please add illustrations on the language models.

Please provide a more in-depth description of corpus data (not all readers are familiar with babel) for each language involved. Please explain the accuracy verification mechanism.

Please provide an in-depth statistical reliability analysis in the experimental section. Please add the analysis of extremities (worst and best cases of recognition). Please add probability density functions.

Please add discussions, linking your results with the ones overviewed in my suggested table format of other languages. Please move future works from conclusions to this section. Conclusions must be just an objectively measurable overview of what you have achieved and why.

Author Response

(The authors gave the same response as above.)

Round 2

Reviewer 2 Report

My comments have been addressed, therefor I recommend accepting this paper.